# Inhibition of Ribosomal RNA Processing 15 Homolog (RRP15) Suppressed Tumor Growth, Invasion and Epithelial to Mesenchymal Transition (EMT) of Colon Cancer

**DOI:** 10.3390/ijms24043528

**Published:** 2023-02-09

**Authors:** Zirong Deng, Yun Xu, Yuchen Cai, Weiling Lin, Libei Zhang, Aoqing Jiang, Yuhang Zhou, Rui Zhao, Heyan Zhao, Zhaoguo Liu, Tingdong Yan

**Affiliations:** Department of Pharmacology, School of Pharmacy, Nantong University, Nantong 226001, China

**Keywords:** ribosomal RNA processing 15 Homolog, colon cancer, tumor growth, invasion, epithelial-mesenchymal transition

## Abstract

Although ribosomal RNA processing 15 Homolog (RRP15) has been implicated in the occurrence of various cancers and is considered a potential target for cancer treatment, its significance in colon cancer (CC) is unclear. Thus, this present study aims to determine RRP15 expression and biological function in CC. The results demonstrated a strong expression of RRP15 in CC compared to normal colon specimens, which was correlated with poorer overall survival (OS) and disease-free survival (DFS) of the patients. Among the nine investigated CC cell lines, RRP15 demonstrated the highest and lowest expression in HCT15 and HCT116 cells, respectively. In vitro assays demonstrated that the knockdown of RRP15 inhibited the growth, colony-forming ability and invasive ability of the CC cells whereas its overexpression enhanced the above oncogenic function. Moreover, subcutaneous tumors in nude mice showed that RRP15 knockdown inhibited the CC growth while its overexpression enhanced their growth. Additionally, the knockdown of RRP15 inhibited the epithelial–mesenchymal transition (EMT), whereas overexpression of RRP15 promoted the EMT process in CC. Collectively, inhibition of RRP15 suppressed tumor growth, invasion and EMT of CC, and might be considered a promising therapeutic target for treating CC.

## 1. Introduction

Latest epidemiological investigations have ranked colon cancer (CC) as the third most common cancer and the second cause of cancer-related deaths worldwide, with increasing trends in incidence and mortality rate in recent years [1]. Although the outcomes of CC patients have achieved marked improvements due to the development of novel therapeutic approaches [2,3], patients’ prognoses remain unsatisfactory. Notably, accumulating evidence has shown that targeted therapy has become an important means for CC treatment [4]; however, drug resistance is the main factor limiting the clinical application of targeted therapy [5], thereby increasing the need for identifying novel biomarkers.

Ribosomal RNA Processing 15 Homolog (RRP15), a nucleolar protein-coding gene located on chromosome 1 [6], was reported to participate in several biological processes, such as nucleolar formation, ribosome biogenesis, embryonic development, etc. [7,8]. As ribosomes are considered macromolecular machines of protein synthesis, their biogenesis is critical for cellular protein synthesis [9] and has been closely associated with cellular growth, differentiation and even tumorigenesis [7]. Interestingly, the correlation between RRP15 aberrant expression and carcinogenesis has been demonstrated in several cancers. Blake et al. reported a significant increase in ribosomal biogenesis in the skin following ultraviolet ray (UVR) exposure, leading to UVR-induced melanoma, which was linked to RRP15 [10]. This suggests that RRP15 mediated the development of UVR-induced melanoma. In addition, Zhao et al. described the functional implications of RRP15 in hepatocarcinogenesis, whereby its high expression in hepatocellular carcinoma (HCC) cell lines and tumors correlated with poorer patient survival outcomes. Moreover, in vitro and in vivo assays showed that RRP15 knockdown repressed HCC proliferation [11]. The mechanistic study found that depletion of RRP15 induced senescence and metabolically shifted glycolytic pentose-phosphate to mitochondrial oxidative phosphorylation, highlighting RRP15 as a promising target for treating HCC. However, the significance of RRP15 in CC remains largely undetermined.

Here, we explored the expression of RRP15 in human clinical CC samples, its association with clinicopathological factors, and biological implications in CC via in vitro and in vivo assays following RRP15 overexpression and knockdown as an attempt to determine its potential relevance as a novel target for CC treatment.

## 2. Results

### 2.1. RRP15 Expression Was Markedly Increased in Human Clinical CC Samples, and Correlated with Poor Prognosis

We investigated RRP15 expression in human clinical CC and adjacent normal colon tissues. Immunohistochemical staining showed a significant increase in RRP15 expression in the CC samples compared with the non-tumoral tissues (Figure 1A), which was further confirmed by Western blotting analysis of eight pairs of collected samples (Figure 1B,C). Next, we explored the relationship between high RRP15 expression and patients’ survival by using GEPIA, which revealed a significant correlation with poorer OS (Figure 1D) and DFS (Figure 1E).

### 2.2. High RRP15 Expression Identified in Colon Tissues of AOM/DSS-Induced Mice CAC Model

Here, we determined RRP15 expression in an AOM/DSS-induced mice CAC model. Consistent with the previous studies [12,13], including ours [14], obvious tumorigenesis and shorter colon length were observed in the AOM/DSS-treated group than in the untreated group (Figure 2A). Interestingly, a significant increase in RRP15 expression was observed in the AOM/DSS-treated group than in the untreated group (Figure 2B). Furthermore, the Western blot also showed higher RRP15 expression in AOM/DSS-treated group than in the untreated group (Figure 2C,D).

### 2.3. RRP15 Protein Expression in CC Cell Lines

Here, we investigated the biological function of RRP15 in CC by evaluating RRP15 expression in nine human CC cell lines, SW480, Colo-205, SW620, HCT15, DLD-1, HCT116, Caco-2, RKO and LOVO. Of all nine cell lines, the Western blot showed that the HCT15 cells demonstrated the highest RRP15 expression while the lowest expression was observed in HCT116 cells (Figure 3A,B). Therefore, HCT15 and HCT116 cells were selected to conduct the following experiments based on the above result.

### 2.4. RRP15 Knockdown Inhibited CC Cell Growth and Colony Formation In Vitro

RRP15 shRNA plasmids were transfected into HCT15 cells for knocking down RRP15, whereas RRP15 pcDNA3.1(+) plasmids were transfected into HCT116 cells for overexpressing RRP15. Western blotting showed that RRP15 shRNA evidently decreased RRP15 expression in HCT15 cells (Figure 4A), and RRP15 pcDNA3.1(+) notably increased RRP15 expression in HCT116 cells (Figure 4B), suggesting that the shRNA or pcDNA3.1(+) plasmids were constructed successfully. RRP15 knockdown repressed HCT15 cells’ proliferation via CCK8 assay (Figure 4C), whereas overexpression of RRP15 enhanced the proliferation of HCT116 cells (Figure 4D). Furthermore, RRP15 depletion inhibited the colony formation of HCT15 cells (Figure 4E,G), whereas overexpression of RRP15 enhanced the colony-forming ability of HCT116 (Figure 4F–H). Collectively, inhibition of RRP15 suppressed the cell growth and colony-forming ability of CC cells.

### 2.5. RRP15 Knockdown Suppressed CC Cells Invasion In Vitro

Transwell assays revealed that RRP15 knockdown notably reduced cell invasion (Figure 5A,B), whereas its overexpression markedly increased cell invasion (Figure 5C,D), indicating that inhibition of RRP15 suppressed the tumor invasion of CC cells.

### 2.6. RRP15 Knockdown Suppressed CC Cells Growth In Vivo

CC xenografts nude mice model were examined to study the effects of RRP15 overexpression or knockdown on tumor growth in vivo. We found that RRP15 overexpression enhanced HCT116 CC xenografts’ growth (Figure 6A,B), tumor weight (Figure 6C) and volume (Figure 6D) compared to the control group. In contrast, the knockdown of RRP15 decreased the tumor weight and volume of HCT15 CC xenografts (Figure 6E–H), suggesting that RRP15 knockdown could inhibit CC cell growth in vivo.

### 2.7. RRP15 Expression Positively Correlated with EMT in Human Clinical CC Samples, and RRP15 Knockdown Inhibited EMT Process of CC Cells In Vitro

EMT has important roles in CC development from initiation to metastasis [15]. Thus, the role of RRP15 in the EMT process of CC was also addressed. The results showed high RRP15 expression with high N-cadherin expression and low E-cadherin expression in the clinical CC samples compared to normal colon tissues (Figure 7A), suggesting a positive correlation between RRP15 expression and EMT process in CC. Western blot assay further showed that RRP15 knockdown increased E-cadherin expressions and decreased N-cadherin expressions in HCT15 cells (Figure 7B,C). On the contrary, RRP15 overexpression decreased E-cadherin expressions and increased the N-cadherin expressions in HCT116 cells (Figure 7D,E). Altogether, these results demonstrated that RRP15 protein promoted EMT in CC.

## 3. Discussion

Recent studies suggested that RRP15 could be a promising therapeutical target for cancer treatment; however, the potential molecular mechanism and clinical significance of RRP15 remains to be determined in CC. In this present study, high RRP15 expression was found both in CC patients and AOM/DSS-induced mice colon tissues. Notably, high RRP15 expression was found correlated with the poor survival rate of CC patients. We also observed that RRP15 knockdown suppressed the proliferation, colony formation and invasion, and notably suppressed tumor growth. In contrast, overexpression of RRP15 promoted CC cell proliferation, colony formation and invasion in vitro, and enhanced tumor growth in vivo. Moreover, RRP15 was involved in EMT regulation in CC, as RRP15 knockdown weakened the EMT whereas RRP15 overexpressed enhanced the EMT. Therefore, we provide evidence showing that RRP15 knockdown suppressed tumor growth, invasion and the EMT of CC. Altogether, this is the first study to evaluate the roles of RRP15 in regulating the biological functions of CC cells.

As a multifunctional protein, RRP15 has been found to play an essential role in ribosome biogenesis. Compelling evidence has shown that RRP15 is involved in many biological processes including cell proliferation, cell cycle progression, nucleolar stress response, apoptosis and checkpoint response [11,16]. This study found that depletion of RRP15 suppressed cell proliferation and cell cycle and increased apoptosis in NIH3T3 cells, whereas RRP15 overexpression increased cell proliferation and cell cycle together with suppression of apoptosis in NIH3T3 cells. This highlights that RRP15 played an important role in the regulation of the biological behavior in NIH3T3 cells [8]. Another study showed that RRP15 depletion inhibited cell proliferation and delayed cell-cycle progression in HeLa cervical cancer or MCF7 breast cancer cells [7]. Zhao et al. found that aberrant high expression of RRP15 in hepatocellular carcinoma (HCC) and high RRP15 expression correlated with poor patient survival. Furthermore, depletion of RRP15 inhibited the tumor growth, induced DNA damage and apoptosis, induced glucose metabolism remodeling and cellular senescence in HCC [11]. Notably, RRP15 was also involved in the occurrence and development of UVR-induced melanoma. This study showed that UVR exposure notably increased the biogenesis of ribosomes in the skin, and RRP15 was found to be the only gene that was remarkably deregulated by UVR. Moreover, the deregulated RRP15 not only correlated with ribosome biogenesis, but also played an essential role in the identification, repair, transcription and splicing of DNA damage [10]. Interestingly, one recent study reported a decreased expression in pancreatic cancer and that RRP15 was associated with cinchonine-induced cell death, while RRP15 knockdown could suppress autophagy and induce apoptosis [16]. In accordance with the published literature, we found that RRP15 participated in the regulation of biological behavior in CC, and affected the cell proliferation, invasion, colony formation and growth. Notably, the aberrant expression of RRP15 was closely linked to the prognosis of CC patients: its high expression led to decreased OS and DFS, highlighting the potential of RRP15 as a prognostic marker in CC patients. In future studies, we would knockout RRP15 in mice to further investigate the regulatory functions of RRP15 in CC in vivo. Thus, our study further expanded the scope of how RRP15 is involved in cancer development except for cervical, breast and liver cancer.

EMT is a highly dynamic process whereby epithelial cells acquire mesenchymal features due to normal or pathological stimuli [17]. EMT leads to the loss of adhesion between tumor cells; therefore, EMT is often associated with carcinogenesis, metastasis, immune evasion and therapeutic resistance [18,19]. The EMT process was found closely linked to the development of CC [20]. The literature has shown that most deaths of CC patients were caused by tumor recurrence, metastasis and drug resistance, and CC stem cells (CCSCs) were found involved in all these processes [21]. Recent studies have found that there is an important correlation between EMT and CCSCs [22]. The acquisition and maintenance of “stemness” of CCSCs is closely related to the EMT. At the same time, CCSCs have EMT characteristics in different microenvironments, thus playing a pivotal role in the tumor occurrence, progression and metastasis [23]. Therefore, targeting EMT could be a promising treatment approach for CC patients. E-cadherin and N-cadherin are two key markers of EMT [24], and the decreased expression of E-cadherin and increased expression of N-cadherin are well-known hallmarks of EMT [25]. In this study, clinical CC samples assay showed that high RRP15 expression was concomitant with decreased E-cadherin and increased N-cadherin expressions, indicating that RRP15 expression was positively correlated with EMT in CC. Furthermore, we assessed the regulatory role of RRP15 in the EMT of CC. Results found that depletion of RRP15 increased E-cadherin and decreased N-cadherin expressions, whereas RRP15 overexpression decreased E-cadherin and increased N-cadherin expressions in CC cells, highlighting that RRP15 regulated tumor invasion through modulation of EMT in CC, confirming the clinical potential of RRP15 as a target for EMT suppression in CC.

Notably, the limitations of this study are as follows: (1) we found that HCT15 cells had poor tumorigenicity while HCT116 cells had good tumorigenicity in subcutaneous transplant tumor; (2) Our study also showed higher RRP15 expressions in AOM/DSS-induced CAC mice, its functions and regulatory mechanisms in AOM/DSS-induced CAC mice urge further clarification. In the future study, RRP15 knockout mice will be further used to study the roles of RRP15 in CC tumorigenesis and AOM/DSS-induced CAC mice; (3) The underlying mechanisms of RRP15 in regulating CC cell proliferation and invasion also require further investigation; (4) As is known to all, CC is prone to liver metastasis, thus whether RRP15 participates in regulating liver metastasis should be clarified. Our future investigations would focus on clarifying the correlation between RRP15 and liver metastasis in CC.

## 4. Materials and Methods

### 4.1. Clinical Specimens

The fresh tissues from treatment-naive CC patients prior to surgical treatment were provided by Doctor Kang Ding from Nanjing Hospital of Chinese Medicine. All the study procedures were approved by the Ethics Committee of Nanjing Hospital of Chinese Medicine on 17 March 2020. (project identification code: KY2020034).

### 4.2. Cell Culture

CC cells, including SW480, Colo-205, SW620, HCT15, DLD-1, HCT116, Caco-2, RKO and LOVO were bought from Shanghai Zhong Qiao Xin Zhou Biotechnology. SW480 and SW620 were cultured in a complete DMEM medium supplemented with 10% FBS (Gibco Grand Island, NY, USA). Colo-205, HCT15 and DLD-1 cells in a complete RPMI-1640 medium supplemented with 10% FBS. Caco-2 and RKO cells in a complete EMEM medium supplemented with 10% FBS. HCT116 cells in a complete Mccoy’s 5A medium supplemented with 10% FBS. LOVO cells in a complete F-12K medium supplemented with 10% FBS. All the experimental procedures described below were repeated thrice unless described otherwise.

### 4.3. Plasmid Construction and Transfection

RRP15 shRNA plasmid was constructed to suppress the RRP15 gene in HCT15 cells, and control shRNA plasmid was used as the negative control. RRP15 pcDNA3.1(+) plasmid was constructed to overexpress the RRP15 gene in HCT116 cells, and control vector plasmid was used as the negative control. All plasmids were designed, synthesized and obtained from Suzhou GenePharma Co., Ltd. (Suzhou, China). Then, transient transfection was carried out with Lipofectamine 2000 (Lot No: 2398587; Invitrogen) according to the protocol. The transfection efficiency was examined via Western blotting.

### 4.4. Cell Count Kit-8 (CCK-8) Assay

First, we seeded and cultured HCT15 or HCT116 CC cells (density, 2 × 10^3^ cells/well) onto a 96-well plate in 100 µL of their corresponding medium at 37 °C for 0 h, 24 h, 48 h and 72 h. Secondly, we added 10 μL of CCK-8 solution (C0043; Beyotime, Shanghai, China) to each well followed by 2 h incubation at similar conditions. Lastly, we measured the cells’ optical density (OD) at 450 nm using the enzyme labeling instrument (SYNERGY H1; BioTek, Winooski, VT, USA).

### 4.5. Colony Formation Assay

CC cells were dispersed into single cells and cultured onto 6-well plates (500 cells per well) in their respective complete medium for two weeks, followed by 15 min methanol fixation of the derived colonies at room temperature and 20 min staining with 0.5% crystal violet. Then, the cells were washed with PBS, dried at room temperature, followed by colony counting under a light microscopy.

### 4.6. Transwell Invasion Assay

First, the matrix gel was diluted with ice-cold serum-free medium and then the filter of the upper chamber was pre-coated with matrix gel. Secondly, 2 × 10^5^ cells were seeded and cultured in a serum-free medium in the upper chamber. Next, we added a complete medium in the lower chamber and left it for 24 h (HCT15 cells) or 48 h (HCT116 cells) in an incubator with 5% CO_2_ at 37 °C. Next, the medium was removed, the invaded cells were stained for 20 min with 0.5% crystal violet, washed with PBS, and assessed and photographed under a microscope.

### 4.7. Immunohistochemical Staining (IHC)

Tissue sections fixed in paraformaldehyde and paraffin were used for IHC following the previously described methodology [26]. Briefly, the slides underwent deparaffinization and rehydration in ethanol-graded solutions and distilled water, submerged for 30 min in 3% H_2_O_2_ in methanol, washed with PBS, followed by 30 min incubation in 10% normal goat serum and washing. Then, they were incubated overnight at 4 °C with the following antibodies: RRP15 (ab121832, 1:300, Cambridge, Cambridgeshire, UK), E-cadherin (ab76319, 1:100) and N-cadherin (ab76011, 1:500). Next, the sections underwent washing three times with PBS, incubation with secondary antibodies, washing again with PBS, developed with a DAB commercial kit (CoWin Biosciences, Taizhou, China), counterstained using hematoxylin and lastly, protein expressions were assessed and photographed under light microscopy at 200× magnification.

### 4.8. Western Blotting

Total protein was extracted and then quantified using a bicinchoninic acid (BCA) protein assay kit. Western blotting assay was performed as previously reported [27]. Briefly, the total proteins were separated using sodium dodecyl sulfate–polyacrylamide gel electrophoresis (SDS-PAGE), transferred using a wet transfer system onto polyvinylidene fluoride (PVDF) membranes, blocked with 5% non-fat milk and incubated at 4 °C overnight in the following primary antibodies: RRP15 (ab121832; Abcam, Cambridge, Cambridgeshire, UK), E-cadherin (ab76319; Abcam) and N-cadherin (ab76011; Abcam). The following day, the PVDF membranes were incubated at room temperature with their respective secondary antibodies, and protein band detection was conducted on an enhanced chemiluminescence system based on an equivalent loading against GAPDH (ab8245; Abcam). Image J was used for densitometry assessment.

### 4.9. CC Xenografts Mice Model

A total of twenty male BALB/C nude mice (four weeks old) were obtained from the Experimental Animal Center of Nantong University (Nantong, China). This step was approved by the Institutional Animal Ethical Committee of Nantong University on 1 July 2022 (project identification code: S20220701-902), and the experiments were conducted following the NIH Guidelines for Care and Use of Laboratory Animals. Briefly, the mice were well-maintained in a room with a 12/12 h day/night cycle with food and water ad libitum during all experimental procedures. They were randomly classified into the following four groups: Control (HCT116 cells); RRP15 pcDNA3.1(+); Control (HCT15 cells); and RRP15 shRNA. Each group had five nude mice. For the Control (HCT116 cells) group, 2 × 10^6^ HCT116 cells were subcutaneously injected into the right flank of nude mice. For the RRP15 pcDNA3.1(+) group, 2 × 10^6^ HCT116 cells (transfected with RRP15 overexpression plasmid) were subcutaneously injected into the right flank of nude mice. For the Control (HCT15 cells) group, 2 × 10^6^ HCT15 cells were injected subcutaneously into the right flank of nude mice. For the RRP15 shRNA group, 2 × 10^6^ HCT15 cells (transfected with RRP15 shRNA plasmid) were subcutaneously injected into the right flank of nude mice. Every two days, a caliper was used to measure the tumor xenograft’s dimensions. After 16 days post-tumor inoculation, the mice were anesthetized by inhaling 3% isoflurane and euthanized by cervical dislocation. The respective tumors were then removed, photographed and weighed. Tumor xenograft volume (mm^3^) was determined by length × width^2^/2.

### 4.10. Establishment of Azoxymethane (AOM)/Dextran sulphate sodium (DSS)-Induced Mice Colitis-Associated Cancer (CAC) Model

To establish the AOM/DSS-induced mice CAC model, the detailed protocol can be referred to in our previous report [14]. The experimental procedures were permitted by the Institutional Animal Ethical Committee of Nantong University on 7 September 2020 (project identification code: S20200907-306) and performed by abiding to the NIH Guidelines for Care and Use of Laboratory Animals. Briefly, the mice were well-maintained in a room with 12/12 h day/night cycle with food and water ad libitum during all experimental procedures. Male C57BL/6 mice were randomly divided into a Control and an AOM/DSS group. The control group had 10 mice, and the AOM/DSS group had 20 mice per group. The AOM/DSS group was administered AOM (10 mg/kg dissolved in physiological saline) intraperitoneally. After 7 days, the mice were given drinking water containing 2.5% DSS for 7 days, then normal drinking water for 14 days, followed by exposure to two 2.5% (*w*/*v*) DSS treatment cycles. Mice from the control group were provided an equivalent volume of normal saline. Following the treatment procedures by week 12, they were sacrificed by cervical dislocation, their colons were excised to measure their length, cut into pieces and then fixed in a 10% formalin buffer solution. The remaining samples were flash-frozen in liquid nitrogen and stored in a −80 °C refrigerator until use for Western blotting.

### 4.11. Gene Correlation Analysis in GEPIA

Gene Expression Profiling Interactive Analysis (GEPIA) (URL: http://gepia.cancer-pku.cn/index.html) (accessed on 10 September 2021) was conducted for overall survival (OS) and disease-free survival (DFS) analysis and curve drawing the basis of the indicated gene expression with log-rank test in 33 cancer types. Correlation analysis for gene was conducted using datasets retrieved from TCGA [28].

### 4.12. Statistical Analyses

These data are shown as presented as means ± standard deviation (SD). Data analyses were conducted using Student’s *t*-test and one-way ANOVA using GraphPad Prism 5 for Windows. *p* < 0.05 was used to determine statistical significance.

## 5. Conclusions

This study showed that RRP15 expression was associated with poor survival in CC patients. Depletion of RRP15 inhibited colony formation, invasion, tumor growth and EMT in CC cells, while RRP15 overexpression facilitated the above biological behavior in CC cells, highlighting the potential clinical significance of targeting RRP15 for treating CC.

## Figures and Tables

**Figure 1 ijms-24-03528-f001:**
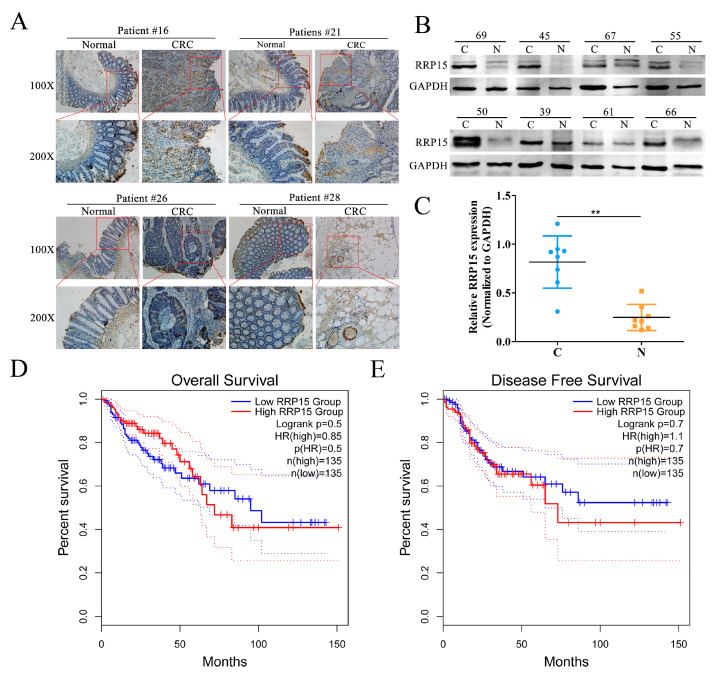
RRP15 expression in human colon cancer (CC) samples and correlation with patient survival. (**A**) Representative images of RRP15 via immunohistochemical (IHC) (scale bar = 100 μm). (**B**) RRP15 protein expression in eight paired CC tissues (marked C) and matched normal tissue samples (marked N) was determined by Western blot. (**C**) Quantitation of Western blot results. The relationship between RRP15 expression and (**D**) the overall survival (OS) and (**E**) disease-free survival (DFS) of CC patients. ** *p* < 0.01 versus the N group, *n* = 8.

**Figure 2 ijms-24-03528-f002:**
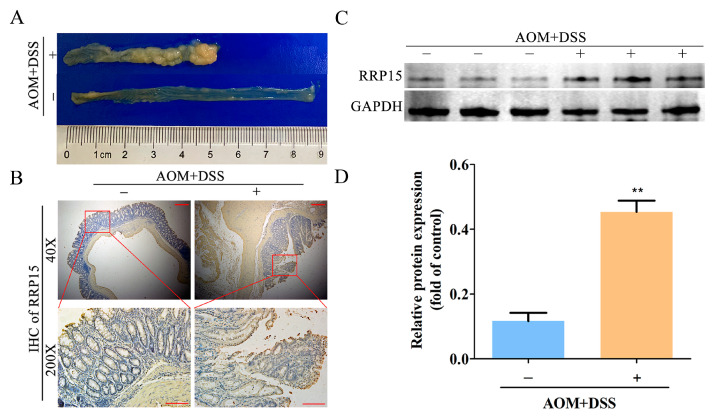
RRP15 expression in the colon tissues of azoxymethane (AOM)/dextran sulphate sodium (DSS)-induced colitis-associated colon cancer (CAC). (**A**) Representative gross morphology of colon tissues in AOM/DSS-induced CAC. (**B**) Representative images of IHC staining of RRP15 in colon tissues of AOM/DSS-induced CAC mice (scale bar = 100 μm). (**C**) Representative blots of RRP15 expression in the colon tissues of AOM/DSS-induced CAC mice. (**D**) Western blot quantitation results expressed as mean ± SD (*n* = 7–10 mice). ** *p* < 0.01 versus the untreated group.

**Figure 3 ijms-24-03528-f003:**
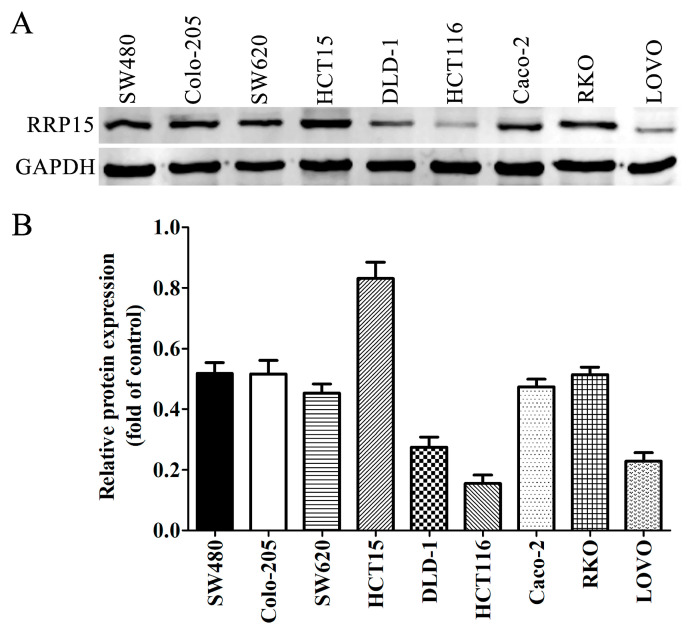
RRP15 protein expression in nine human CC cell lines (SW480, Colo-205, SW620, HCT15, DLD-1, HCT116, Caco-2, RKO and LOVO) via Western blotting. (**A**) RRP15 protein expression in the CC cell lines. (**B**) Western blot quantitation results, *n* = 3.

**Figure 4 ijms-24-03528-f004:**
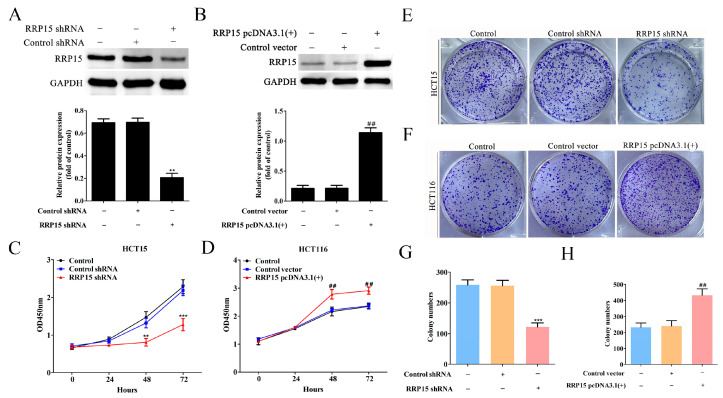
The implication of RRP15 knockdown or overexpression on the cellular proliferation and colony formation of CC cells. (**A**) Western blot analysis of RRP15 expression level in HCT15 cells after RRP15 shRNA plasmid transfection. (**B**) Western blot analysis of RRP15 expression level in HCT116 cells after RRP15 pcDNA3.1(+) plasmid transfection. Effects of RRP15 knockdown on the proliferation of (**C**) HCT15 cells and (**D**) HCT116 cells via CCK8. Effects of RRP15 knockdown on the colony-forming ability of (**E**) HCT15 cells and (**F**) HCT116 cells. (**G**,**H**) Quantitation of the colony numbers. ** *p* < 0.01 and *** *p* < 0.001 versus the control shRNA group; ^##^
*p* < 0.01 versus the control vector group, *n* = 3.

**Figure 5 ijms-24-03528-f005:**
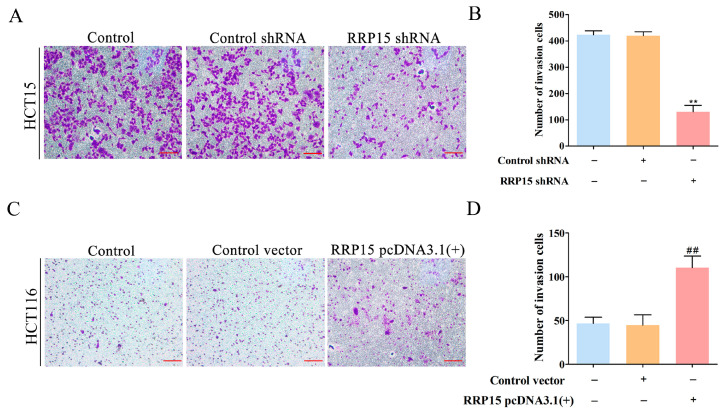
Effects of RRP15 knockdown or overexpression on CC cell invasion. (**A**) Effects of RRP15 knockdown on HCT15 cell invasion (scale bar = 100 μm). (**B**) Quantitation of the number of invasion cells. (**C**) Effects of RRP15 overexpression on HCT116 cell invasion (scale bar = 100 μm). (**D**) Quantitation of the number of invasion cells. ** *p* < 0.01 versus the control shRNA group; ^##^ *p* < 0.01 versus the control vector group, *n* = 3.

**Figure 6 ijms-24-03528-f006:**
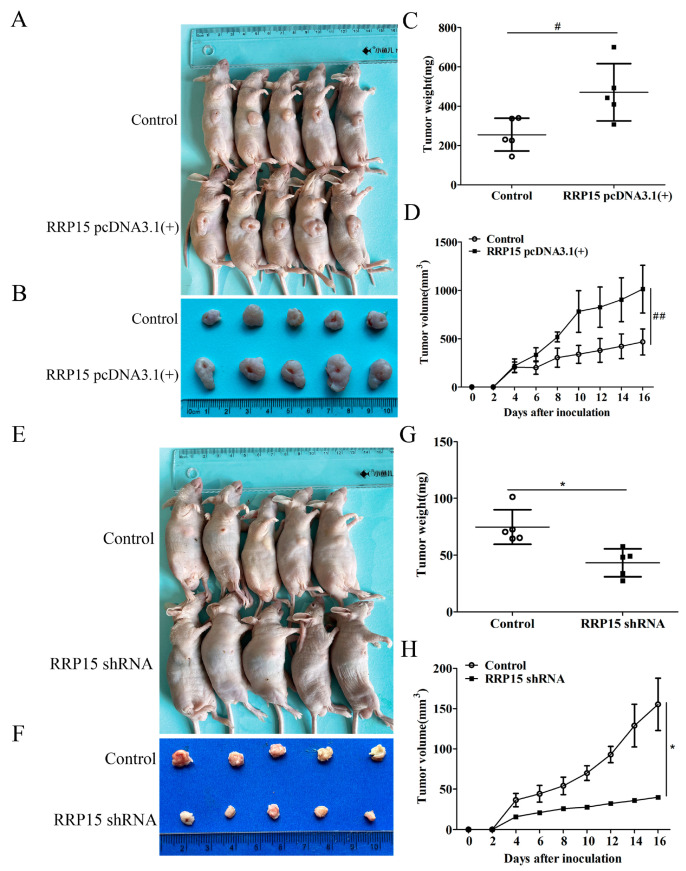
Effects of RRP15 knockdown or overexpression on tumor growth of CC cells. (**A**) Representative images of tumor-bearing nude mice subcutaneously injected with HCT116 cells transfected with RRP15 pcDNA3.1(+) plasmid. (**B**) Images for tumor xenografts from nude mice. (**C**) Tumor weight and (**D**) volume. (**E**) Representative images of tumor-bearing nude mice that were subcutaneously injected with HCT15 cells transfected with RRP15 shRNA plasmid. (**F**) Images for tumor xenografts from nude mice were shown. (**G**) Tumor weight and (H) volume. ^#^ *p* < 0.05 and ^##^ *p* < 0.01 versus the control or control vector group; * *p* < 0.05 versus the control or control shRNA group, *n* = 5.

**Figure 7 ijms-24-03528-f007:**
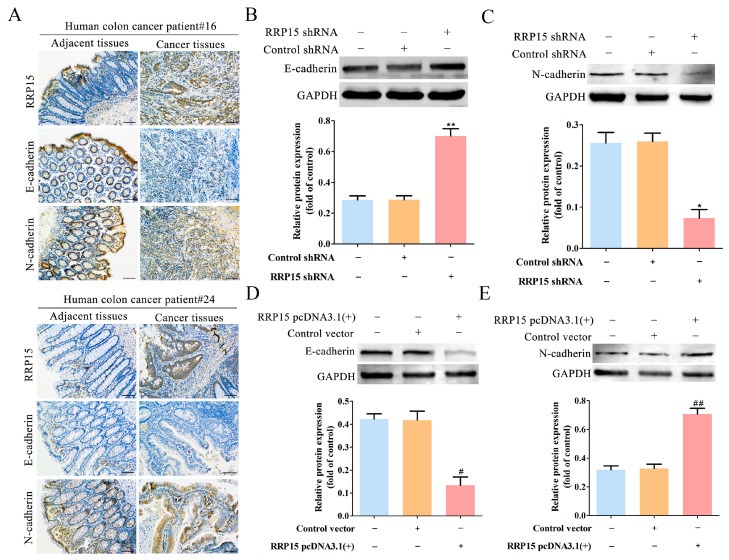
Effects of RRP15 knockdown or overexpression on epithelial-mesenchymal transition (EMT) of CC cells. (**A**) Representative images of IHC staining of RRP15, E-cadherin and N-cadherin in CC tissues and paired adjacent tissues (scale bar = 100 μm). (**B**) Effects of RRP15 knockdown on the expression of E-cadherin in HCT15 cells examined by Western blot. Representative blots via densitometry. (**C**) Effects of RRP15 knockdown on the expression of N-cadherin in HCT15 cells via Western blot. Representative blots are shown with densitometry (*n* = 3). (**D**) The effect of RRP15 overexpression on expression of E-cadherin in HCT116 cells was examined by Western blot. Representative blots are shown with densitometry. (**E**) Effects of RRP15 overexpression on the N-cadherin expression in HCT116 cells by Western blot. Representative blots showed under densitometry. * *p* < 0.05 and ** *p* < 0.01 versus the control shRNA group; ^#^ *p* < 0.05 and ^##^ *p* < 0.01 versus the control vector group, *n* = 3.

## Data Availability

All the data are available within the article.

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
