# Peer review of "Inhibition of Ribosomal RNA Processing 15 Homolog (RRP15) Suppressed Tumor Growth, Invasion and Epithelial to Mesenchymal Transition (EMT) of Colon Cancer"

_ijms, 2023, doi:10.3390/ijms24043528_

Round 1

Reviewer 1 Report

Comments and Suggestions to the Authors.

MS ID: ijms-2151812

Article.

Title: Inhibition of ribosomal RNA Processing 15 homolog (RRP15) suppressed tumor growth, invasion and epithelial to mesen-chymal transition (EMT) in human colon cancer.

Authors: Zirong Deng et al.

In this manuscript “Inhibition of ribosomal RNA Processing 15 homolog (RRP15) suppressed tumor growth, invasion and epithelial to mesenchymal transition (EMT) in human colon cancer " by Zirong Deng et al. Authors presented their data very well with significant effects of how RRP15 expression in CRC promotes tumor growth, invasion and EMT by which they claiming RRP15 can be a promising therapeutic target for CRC treatment in the future direction.

The article is very well presented and well-written. The topic is original and relevant to the field. I found the conclusion is in line with the evidence and arguments presented. The manuscript is interesting, however, it can be improved and strengthened by addressing the following few minor issues-

1.    In figure-1 authors displayed the expression correlation of RRP15 in cancer vs non cancer tissues of the CRC patent samples using western blotting technique, in relation with the loading control few samples shows no difference in expression, please quantify and do statistical test to show the significant differences. Also please present the western blots with lighter background as most of the western data shown in this paper is same poor quality. Hence, please improve the blots to show the clear differences.

2.    I suggest author to show the RRP15 relation in CRC patents underwent chemotherapy and their survival outcomes if has some significant correlation will strengthen the fugure-1.

3.    Scale bars were missing/not visible in Figure-2B; Figure-5A and C please check.

4.    Figure-7 can be improved with standard EMT transcription markers expression along with E-Cadherin and N-Cadherin.

Please address and justify the above discussed points.

With these recommended changes addressed by the authors, this MS can be considered for publication in IJMS.

Good work.

Author Response

Point 1: In figure-1 authors displayed the expression correlation of RRP15 in cancer vs non cancer tissues of the CRC patent samples using western blotting technique, in relation with the loading control few samples shows no difference in expression, please quantify and do statistical test to show the significant differences. Also please present the western blots with lighter background as most of the western data shown in this paper is same poor quality. Hence, please improve the blots to show the clear differences.

Response 1: As the reviewer suggested that we should quantify and do statistical test to show the significant differences, we have quantify and do statistical test according to the reviewer’s suggestion. The detail revision can be seen in Figure 1C in the revised manuscript. Besides, as the reviewer suggested that we should present the western blots with lighter background as most of the western data shown in this paper is same poor quality, we have processed the western blots and present the western blots with lighter background in Figure 1B, Figure 2C, Figure 3A, Figure 4A, B, Figure 7B, C, D, E in the revised manuscript.

Point 2: I suggest author to show the RRP15 relation in CRC patents underwent chemotherapy and their survival outcomes if has some significant correlation will strengthen the fugure-1.

Response 2: As the reviewer suggested that we show the RRP15 relation in CRC patents underwent chemotherapy and their survival outcomes, however, all collected colon cancer tissues were from the patients received no chemo- or radio-therapy before surgery. We are very sorry that we have no samples that CRC patients underwent chemotherapy. In the future study, we will try to collect the clinical samples on CRC patients underwent chemotherapy, and further analysis the correlation between RRP15 expression and survival outcomes of CRC patients. We hope the reviewer can understand us and can satified with our response. Thanks again for the reviewer’s good suggestion.

Point 3: Scale bars were missing/not visible in Figure-2B; Figure-5A and C please check.

Response 3: It is very true as the reviewer mentioned that the scale bars were missing/not visible in Figure-2B; Figure-5A and C, we have added the scale bars, and the detail revision can be seen in Figure-2B; Figure-5A and C in the revised manuscript.

Point 4: Figure-7 can be improved with standard EMT transcription markers expression along with E-Cadherin and N-Cadherin.

Response 4: As the reviewer suggested that Figure 7 can be improved with standard EMT transcription markers expression along with E-Cadherin and N-Cadherin, however, during the study, we only examined the protein expression of E-Cadherin and N-Cadherin, the other standard EMT transcription marker were not studied. We also realized that standard EMT transcription markers could obvious improve Figure 7, however, due to the time limitation (10 days), we have no enough time to conduct further study, so we will study the tandard EMT transcription markers in our future study, we hope the reviewer can satisfied with our response. Thanks very much for the reviewer’s good suggestion.

Reviewer 2 Report

Manuscript titled “Inhibition of ribosomal RNA Processing 15 homolog (RRP15) suppressed tumor growth, invasion and epithelial to mesenchymal transition (EMT) in human colon cancer” highlights the role of RRP15 in tumorigenesis and metastasis with strong evidences comprises of clinical as well as in-vivo and in-vitro studies. Manuscript showcases good quality data and may be accepted after working on the following comments.

Suggestions/Comments:

Title: The study population is not purely human colon cancer therefore, I would suggest that from title “in human colon cancer” may be taken off.

Abstract:

1.     Line 20: Although CRC is also known as colon cancer, but abbreviation “CRC” is ideally used for the term “Colorectal cancer”. Similar pattern is followed in introduction section too.

2.     Line 24-25: “Noteworthy, RRP15 expression was also notably increased in azoxymethane/dextran sodium sulfate (AOM/DSS)-treated group compared to the untreated group”. Study model is not clear ?”

Introduction:

1.     Line 45-46: Grammatical error in sentence construction. Kindly rephrase the sentence.

2.     I feel that paragraph on EMT is not well-connected with previous paragraph on RRP15. Authors can improve this by either removing EMT in the introduction and incorporating in discussion portion or relating EMT to RRP15.

Material and Methods:

1.     Although in-vitro studies are explaining basics of experiment, I would like to advise to write in-depth procedure of assays like CCK-8 assay, colony formation assay, trans-well invasion assay etc. for more clarity and distinct understanding. 

2.     IHC and western blotting protocols need to be elaborated more for better understanding.

3.     Line 111-112: Author can elaborate more on details of each plasmid construction in terms of which plasmid was constructed to overexpress/suppress the particular gene.

4.     Line 116:  Error in sentence construction. Kindly rephrase the sentence.

5.     Line 118: 2 × 103 or 2 × 103 ?

6.     Line119-120: Error in sentence construction

7.     Line 122-123: Error in sentence construction. Meaning of this line is not conveyed.

8.     Line 130: 2×105 cells or 2 × 105 ?

9.     Line 166: Need to use scientific term like cervical dislocation instead of “breaking the neck”

10.  Section 2.9: Kindly elaborate on general physiological parameters of animals, housing conditions, food and water intake etc.

Results:

1.     Section 3.1: Kindly provide high quality image (more than 300dpi) for graphical representation of overall survival and disease-free survival.

2.     Section 3.7: Title can be modified considering the representation of clinical data also and not only in-vitro results.

Discussion:

1.     Discussion section needs to be elaborated more properly in a flow similar to results section. Kindly avoid repetition of results in discussion portion. Literature support is inadequate in discussion section. Author may try to add more related literature to correlate the current findings.

2.     Line 322-323: Need to improve the sentence construction. Meaning of this line is not conveyed properly.

Author Response

Point 1: Title: The study population is not purely human colon cancer therefore, I would suggest that from title “in human colon cancer” may be taken off.

Response 1: It is really true as the reviewer mentioned that the study population is not purely human colon cancer, and suggested that “in human colon cancer” may be taken off. We have corrected the title, and the detail revision was marked in red can be seen in the reviased manuscript.

Point 2: Line 20: Although CRC is also known as colon cancer, but abbreviation “CRC” is ideally used for the term “Colorectal cancer”. Similar pattern is followed in introduction section too.

Response 2: As the reviewer mentioned that abbreviation “CRC” is ideally used for the term “Colorectal cancer”, and we should correct it. We have corrected the description to “CC” across the whole manuscript, the detail revision was marked in red can be seen in the reviased manuscript.

Point 3: Line 24-25: “Noteworthy, RRP15 expression was also notably increased in azoxymethane/dextran sodium sulfate (AOM/DSS)-treated group compared to the untreated group”. Study model is not clear ?”

Response 3: It is very true as the reviewer mentioned that the study model is not clear in Line 24-25, we have corrected the description, the detail revision was marked in red can be seen in the reviased manuscript.

Point 4: Line 45-46: Grammatical error in sentence construction. Kindly rephrase the sentence.

Response 4: As the reviewer mentioned that grammatical error in sentence construction in Line 45-46, we have rephrased the sentence construction, and the detail revision was marked in red can be seen in the reviased manuscript.

Point 5: I feel that paragraph on EMT is not well-connected with previous paragraph on RRP15. Authors can improve this by either removing EMT in the introduction and incorporating in discussion portion or relating EMT to RRP15.

Response 5: As the reviewer suggested that we can improve the paragraph on EMT by either removing EMT in the introduction and incorporating in discussion portion or relating EMT to RRP15, we have removed EMT in the introduction and incorporating in discussion portion or relating EMT to RRP15, and the detail revision was marked in red can be seen in the reviased manuscript.

Point 6: Although in-vitro studies are explaining basics of experiment, I would like to advise to write in-depth procedure of assays like CCK-8 assay, colony formation assay, trans-well invasion assay etc. for more clarity and distinct understanding.

Response 6: As the reviewer suggested that we could write in-depth procedure of assays like CCK-8 assay, colony formation assay, trans-well invasion assay, we have rewritten the procedure of CCK-8 assay, colony formation assay, trans-well invasion assay, and the detail revision was marked in red can be seen in the reviased manuscript.

Point 7: IHC and western blotting protocols need to be elaborated more for better understanding.

Response 7: It is very true as the reviewer suggested that IHC and western blotting protocols need to be elaborated more for better understanding. We have rewritten the protocols on IHC and western blotting, and the detail revision was marked in red can be seen in the reviased manuscript.

Point 8: Line 111-112: Author can elaborate more on details of each plasmid construction in terms of which plasmid was constructed to overexpress/suppress the particular gene.

Response 8: As the reviewer suggested that we could elaborate more on details of each plasmid construction in terms of which plasmid was constructed to overexpress/suppress the particular gene, we have done it according to the reviewer’s good suggestion, and the detail revision was marked in red can be seen in the reviased manuscript.

Point 9: Line 116: Error in sentence construction. Kindly rephrase the sentence.

Response 9: It is very true as the reviewer mentioned that error in sentence construction in Line 116, we have rephrased the sentence, and the detail revision was marked in red can be seen in the reviased manuscript.

Point 10: Line 118: 2 × 103 or 2 × 103 ?

Response 10: We are very sorry for our negligence, we have corrected the mistake, and the detail revision was marked in red can be seen in the reviased manuscript.

Point 11: Line119-120: Error in sentence construction

Response 11: It is very true as the reviewer mentioned that error in sentence construction in Line 119-120, we have rephrased the sentence, and the detail revision was marked in red can be seen in the reviased manuscript.

Point 12: Line 122-123: Error in sentence construction. Meaning of this line is not conveyed.

Response 12: It is very true as the reviewer mentioned that error in sentence construction in Line 122-123, we have rephrased the sentence, and the detail revision was marked in red can be seen in the reviased manuscript.

Point 13: Line 130: 2×105 cells or 2 × 105 ?

Response 13: We are very sorry for our negligence, we have corrected the mistake, and the detail revision was marked in red can be seen in the reviased manuscript.

Point 14: Line 166: Need to use scientific term like cervical dislocation instead of “breaking the neck”

Response 14: It is very true as the reviewer mentioned that we need to use scientific term like cervical dislocation instead of “breaking the neck”, we have corrected the description according to the reviewer’s suggestion, and the detail revision was marked in red can be seen in the reviased manuscript.

Point 15: Section 2.9: Kindly elaborate on general physiological parameters of animals, housing conditions, food and water intake etc.

Response 15: As the reviewer suggested that general physiological parameters of animals, housing conditions, food and water intake should be elaborated, we have added the related information, and the detail revision was marked in red can be seen in the reviased manuscript.

Point 16: Section 3.1: Kindly provide high quality image (more than 300dpi) for graphical representation of overall survival and disease-free survival.

Response 16: As the reviewer suggested that high quality image (more than 300dpi) for graphical representation of overall survival and disease-free survival should be provided, we have provided the high quality image (more than 300dpi) for graphical representation of overall survival and disease-free survival in Figure 1 in the revised manuscript.

Point 17: Section 3.7: Title can be modified considering the representation of clinical data also and not only in-vitro results.

Response 17: It is really true as the reviewer suggested that Title can be modified considering the representation of clinical data also and not only in-vitro results, we have modified the title of section 3.7, and the detail revision was marked in red can be seen in the reviased manuscript.

Point 18: Discussion section needs to be elaborated more properly in a flow similar to results section. Kindly avoid repetition of results in discussion portion. Literature support is inadequate in discussion section. Author may try to add more related literature to correlate the current findings.

Response 18: It is really true as the reviewer suggested that we should add more related literature to correlate the current findings. We have added the related literature to correlate the current findings, and the detail revision was marked in red can be seen in the reviased manuscript.

Point 19: Line 322-323: Need to improve the sentence construction. Meaning of this line is not conveyed properly.

Response 19: It is very true as the reviewer mentioned that error in sentence construction in Line 322-323, we have improved the sentence construction, and the detail revision was marked in red can be seen in the reviased manuscript.

Round 2

Reviewer 2 Report

Thankyou for incorporating suggestions. I would recommend manuscript for publication.